# Population birth outcomes in 2020 and experiences of expectant mothers during the COVID-19 pandemic: A 'born in Wales' mixed methods study using routine data

Hope Jones[1]*, Mike Seaborne[1], Laura Cowley[1,2], David Odd[3], Shantini Paranjothy[4], Ashley Akbari[5], Sinead Brophy[1]

1 National Centre for Population Health and Wellbeing Research, Swansea University Medical School, Swansea, Wales, United Kingdom, 2 Research and Evaluation Division, Knowledge Directorate, Public Health Wales, Cardiff, Wales, United Kingdom, 3 Division of Population Medicine, School of Medicine, Cardiff University, Cardiff, Wales, United Kingdom, 4 Centre for Health Data Science, University of Aberdeen, Aberdeen, Scotland, United Kingdom, 5 Population Data Science, Health Data Research UK, Swansea University Medical School, Swansea, Wales, United Kingdom

* h.e.jones@swansea.ac.uk

**Data Availability Statement:** The datasets supporting conclusions from this article are

## Abstract

### Background

Pregnancy can be a stressful time and the COVID-19 pandemic has affected all aspects of life. This study aims to investigate the pandemic impact on pregnancy experience, rates of primary childhood immunisations and the differences in birth outcomes in during 2020 to those of previous years.

### Methods

Self-reported pregnancy experience: 215 expectant mothers (aged 16+) in Wales completed an online survey about their experiences of pregnancy during the pandemic. The qualitative survey data was analysed using codebook thematic analysis. Population-level birth outcomes in Wales: Stillbirths, prematurity, birth weight and Caesarean section births before (2016–2019) and during (2020) the pandemic were compared using anonymised individual-level, population-scale routine data held in the Secure Anonymised Information Linkage (SAIL) Databank. Uptake of the first three scheduled primary childhood immunisations were compared between 2019 and 2020.

### Findings

The pandemic had a negative impact on the mental health of 71% of survey respondents, who reported anxiety, stress and loneliness; this was associated with attending scans without their partner, giving birth alone, and minimal contact with midwives. There was no significant difference in annual outcomes including gestation and birth weight, stillbirths, and Caesarean sections for infants born in 2020 compared to 2016–2019. There was an increase in late term births (≥42 weeks gestation) during the first lockdown (OR: 1.28, p =

available via the Secure Anonymised Information Linkage (SAIL) databank, which is part of the national e-health records infrastructure for Wales. The supporting information document lists the data sources from the SAIL databank. For further information on the SAIL databank and enquiries in how to access the data, please visit the SAIL website (http://www.saildatabank.com). The HDRUK Innovation Gateway (www. healthdatagateway.org) holds all the datasets and provides information on how to access them. It is a TRE trusted third party which enables access to the data. For more information about gaining access to SAIL please visit saildatabank.com/application-process/. The data from the survey is being uploaded into SAIL when we get to 500 responses so it will be accessible through SAIL in the same was as mentioned above. The findings can be replicated in their entirety by directly obtaining the data from SAIL and following the protocol in the methods section. The authors did not have any special access privileges that others would not have.

**Funding:** The National Centre for Population Health and Wellbeing Research (NCPHWR) receives funding from Health Care Research Wales which supports this research. This research has been supported by the ADR Wales programme of work (https://www.adruk.org/about-us/our-partnership/adr-wales/). The ADR Wales programme of work is aligned to the priority themes as identified in the Welsh Government's national strategy: Prosperity for All. ADR Wales brings together data science experts at Swansea University Medical School, staff from the Wales Institute of Social and Economic Research, Data and Methods (WISERD) at Cardiff University and specialist teams within the Welsh Government to develop new evidence which supports Prosperity for All by using the SAIL Databank at Swansea University, to link and analyse anonymised data. ADR Wales is part of the Economic and Social Research Council (part of UK Research and Innovation) funded ADR UK (grant ES/S007393/1). This work was supported by Health Data Research UK, which receives its funding from HDR UK Ltd (HDR-9006) (https://www.hdruk.ac.uk/) funded by the UK Medical Research Council, Engineering and Physical Sciences Research Council, Economic and Social Research Council, Department of Health and Social Care (England), Chief Scientist Office of the Scottish Government Health and Social Care Directorates, Health and Social Care Research and Development Division (Welsh Government), Public Health Agency (Northern Ireland), British Heart Foundation (BHF) and the Wellcome Trust. This work was supported by the National Core Studies,

0.019) and a decrease in moderate to late preterm births (32–36 weeks gestation) during the second lockdown (OR: 0.74, p = 0.001). Fewer babies were born in 2020 (N = 29,031) compared to 2016–2019 (average N = 32,582). All babies received their immunisations in 2020, but there were minor delays in the timings of immunisations. Those due at 8-weeks were 8% less likely to be on time (within 28-days) and at 16-weeks, they were 19% less likely to be on time.

## Interpretation

Whilst the pandemic had a negative impact on mothers' experiences of pregnancy. Population-level data suggests that this did not translate to adverse birth outcomes for babies born during the pandemic.

## Introduction

The prenatal period is marked by pronounced physiological and psychosocial changes, and previous work has shown that general anxiety, pregnancy-related anxiety, and psychosocial stress are common in pregnant women [1–6]. Maternal stress and anxiety during pregnancy is also known to be associated with adverse neonatal outcomes while stress is associated with adverse obstetric outcomes [7–9]. Specifically, anxiety and stress during pregnancy have been associated with premature birth and low birth weight [8, 10, 11], which are in turn associated with increased risk of neurodevelopmental and respiratory complications [12, 13], and increased risk of infant mortality [14, 15]. Stress in pregnancy has also been associated with stillbirth [16] and contributes to a higher likelihood of unplanned caesarean delivery and prolonged labour duration via the use of analgesia [17, 18]. The relationship between prenatal anxiety and obstetric outcomes is not clear cut [19, 20] and warrants further exploration.

After the World Health Organization (WHO) declared the novel coronavirus (COVID-19) outbreak a global pandemic on 11 March 2020 [21], public health measures and non-pharmaceutical interventions (i.e. social distancing, lockdowns, self-isolation and shielding) were implemented across the UK in order to control the spread of the virus. These restrictions led to major changes for the delivery of primary and secondary care services, including changes in how antenatal, intra-partum and post-natal care was provided. In Wales, partners were unable to attend antenatal or ultrasound appointments, or to be present during labour and delivery [22]. Furthermore, pregnant women may have lacked social support from their friends, family, and community due to social distancing and lockdown measures, and routine contact with health visitors during the postnatal period was also disrupted. Many may have taken additional precautions to avoid contact with others, as pregnant women are considered a high-risk population [23].

Experiencing pregnancy during a pandemic potentially adds a unique element of additional stress for expectant mothers. Pregnant women may be particularly susceptible to the adverse, indirect effects of the COVID-19 pandemic and associated restrictions [24]. Pandemic-related stress and adversity may trigger or exacerbate common prenatal mental health conditions. An increasing number of studies worldwide have reported heightened levels of stress and anxiety among pregnant women because of the pandemic and the broad changes to antenatal, intra-partum and postnatal care pathways [25–28]. However, studies examining neonatal and obstetric outcomes have thus far produced mixed results, with some, but not all, reporting a

an initiative funded by UKRI, NIHR and the Health and Safety Executive (https://www.hdruk.ac.uk/covid-19/covid-19-national-core-studies/). The COVID-19 Longitudinal Health and Wellbeing National Core Study was funded by the Medical Research Council (MC_PC_20030). Grant number for National Centre for Population Health and Wellbeing Research (NCPHWR) AMS103836 The funders had no role in study design, data collection and analysis, decision to publish, or preparation of the manuscript.

**Competing interests:** The authors have declared that no competing interests exist.

higher incidence of stillbirths [29, 30], and others reporting decreases in premature births [31–33].

While a number of studies have assessed women's experiences of pregnancy during the COVID-19 pandemic [34–36], studies conducted in Wales are lacking. In addition, there are a lack of studies examining experiences of pregnancy during the pandemic in combination with national data on objective measures of obstetric, neonatal and infant outcomes including immunisation uptake which may potentially indicate disruption to usual access to healthcare.

This study aims to examine (a) women's experience of pregnancy during the coronavirus pandemic and b) if there was any change in population birth outcomes including stillbirths, mortality, prematurity, birth weight, rates of Caesarean sections (C-sections) and primary childhood immunisations before and during 2020.

## Materials and methods

### Study design

There were two parts to this study: 1) An online survey with a sub-group of expectant mothers about their experiences of pregnancy during the COVID-19 pandemic and 2) Analysis of data which is routinely collected about pregnancy and birth outcomes in Wales, before and during the pandemic.

**Survey for expectant mothers.**   Expectant mothers aged 16+ living in Wales during the COVID-19 pandemic were invited to complete an online survey via social media advertising. Online consent was taken prior to completion. The survey took approximately 20 minutes to complete. Closed questions were used to ascertain information about participants' demographic characteristics, and whether they had experienced periods of stress, anxiety, or stressful life events during their pregnancy. We used the stress questions from the Pregnancy Risk Assessment Monitoring System (PRAMS) [37]. We also used the Patients Health Questionnaire (PHQ-9) [38] and the General Anxiety Disorder (GAD-7) [39] to assess anxiety and depression.

Quantitative survey data were summarised using descriptive statistics. Codebook thematic analysis [40] was used to generate themes from an open-ended question on the survey: 'How would you describe your experience of this pregnancy (support from midwife, how do you feel about being pregnant)?' Thematic analysis identifies and describes patterns across data [41]. 'Codebook' approaches use a structured coding framework to develop and document the analysis [40, 42]. Analysis involved six phases 1) data familiarisation and writing familiarisation notes 2) systematic data coding 3) generating initial themes from coded and collated data 4) developing and reviewing themes 5) refining, defining, and naming themes and 6) writing the report. All data were independently analysed by HJ and LC, who then discussed their findings. This was to ensure that important concepts within the data were not missed, and to achieve a richer understanding of the data through multiple perspectives [40].

### Total population linked data

A retrospective cohort of babies born in 2016 through to the end of 2020 was created by using linked, electronic health record (EHR) data sources available within the Secure Anonymised Information Linkage (SAIL) Databank [43–47]. The SAIL Databank is a privacy-protecting trusted research environment (TRE) that holds anonymised, individual-level, population-scale linkable data sources from ~5-million of the living and deceased population of Wales, that enables longitudinal retrospective and prospective follow-up using health and social care data.

The records are anonymised using a split-file approach; the demographic and clinical data are divided and sent to a trusted third party, Digital Health and Care Wales (DHCW) where a

unique linking field is applied, removing any identifiers. This allows the files to be recombined later and for data to be linked across data sources.

The data sources used for this study included: National Community Child Health (NCCH), Annual District Death Extract (ADDE) from the Office for National Statistics (ONS) mortality register, Patient Episode Dataset for Wales (PEDW), Welsh Demographic Service Dataset (WDSD), Welsh Longitudinal General Practice (WLGP), and COVID-19 Polymerase chain reaction (PCR) testing data (PATD). See S1 Table.

NCCH data were linked to primary and secondary care data sources and compared for babies born between January and December 2020 and those born before (2016–19). Birth outcomes were stillbirths, gestational age at birth, rate of C-sections, and mortality. Covariates affecting outcomes and relating to possible pandemic differences included: residing in rural or urban areas, ethnicity and deprivation level. Data definitions for birth outcomes can be found in S2 Table. Where gestational age was missing (1327 [0.8%] cases), birth weights were consistent with term births. Clinical experience of the author also suggests that gestational ages are often better recorded for non-term births. These missing values were therefore assigned a 40-weeks gestation and the category of 'term'. Missing data was otherwise treated as missing.

Descriptive statistics (percentage, mean and standard deviation) were used to calculate comparative outcomes of annual and monthly incidence of infant characteristics. Comparison was made between births in 2020, and the reference population (birth 2016–2019). Birth outcomes evaluated were stillbirths, gestational age at birth, birth weight, rate of C-sections, and mortality.

Odds and odds ratios (OR) were also calculated to compare pre-pandemic outcomes with outcomes in the 2020 pandemic epoch, using unconditional maximum likelihood estimations. 95% confidence intervals were used with p-values of less than 0.05 representative of statistical significance.

Outcomes were also stratified by rural and urban populations; and areas of differing Welsh Index of Multiple Deprivation (WIMD) quintiles version 2019. WIMD ranks small areas of Wales based on eight separate domains. These include: income, employment, health, education, access to services, housing, community safety and physical environment [48]. Both rural and urban populations were linked at an individual-level based on the anonymised residence information in WDSD using the Lower-layer Super Output Area (LSOA) version 2011.

A post-hoc join point Poisson regression analysis was performed on preterm mortality rates to test if there was a temporal relationship. It was used to refine our understanding of the relationship which otherwise only compared rates in 2020 to the mean rates of 2016–2019.

Comparisons of routine childhood immunisations [49–51] were made between 2019 and 2020 for doses due at 8-, 12- and 16-weeks chronological age. No analysis of immunisation data prior to 2019 was undertaken because of variation of their codes over time.

There were some inconsistencies around dates in which immunisations were administered and likely some missing data where not all immunisations due at a given time were recorded. If at least one of the due immunisations were given, it was assumed all others due at the same time were given whether recorded at the same date/time or not.

Immunisations were considered 'on time' if given within 28-days of their due date as guidance issued to healthcare workers in Wales scheduled immunisation by age in months rather than weeks [52–54]. Second and third doses were on time if administered at 28- and 56-days respectively of the first dose these doses were also allowed to be up to 28-days after these dates.

Information about the software used for cohort selection and analysis can be found in the data in S1 Text.

## Ethical approval

This study was approved by the SAIL Databank independent Information Governance Review Panel (project number 0916, Wales Electronic Cohort for Children Phase 4). The qualitative and survey aspects of the study were approved by HRA and Health and Care Research Wales (HCRW) REC reference: 21/NW/0156.

## Results

### Survey results

The survey received 215 responses between 7[th] September 2020 and 1[st] April 2021. Of the respondents, 203 (94%) were from a white ethnic background and average age was 32 years (See Table 1 below). 45% of expectant mothers responded 'yes' to whether they had periods of bad stress or stressful life events during their pregnancy (N = 96). They were asked on a scale of 0 to 10, how stressful was this time (0 is not at all, 10 is overwhelming). The mean score was 7.43. 26% (N = 26) said this stressful event was related to coronavirus. 25% (N = 24) had someone close with a serious illness. 19% (N = 18) experienced serious relationship difficulties with their husband or partner. During this time 69% (N = 66) had someone who could support them emotional or financially. Expectant mothers who had periods of stress during their pregnancy reported higher anxiety levels than those who had no periods of stress. 84% of expectant mothers who had experienced periods of stress reported feeling nervous, anxious or on edge from several days a week to nearly every day compared to 48% of mothers who reported no stress. This anxiety was experienced 'not at all' by 52% of mothers who reported no periods of stress, compared to only 12% of mothers who had periods of stress.

Three key themes were developed from the qualitative data: (1) Perception of the severity of the COVID-19 pandemic, (2) difference to regular appointments and delivery and (3) support from midwives. A coding framework detailing the themes, subthemes and definitions is provided in S3 Table. Some expectant mothers described their experiences of pregnancy during the COVID-19 pandemic in positive terms, such as 'good', 'great' or 'excellent', and reported that they felt 'happy', 'calm', 'excited' or 'ecstatic'. However, 71% of expectant mothers described their experiences as being 'poor', 'awful' or 'terrible', and reported feeling 'stressed', 'uncertain', 'uninformed', 'isolated', 'anxious' and 'overwhelmed'. Others still reported mixed feelings, stating that they were both happy to be pregnant yet anxious about the impact of COVID-19 and associated restrictions on their health and wellbeing and that of their unborn baby. In terms of perception of the pandemic, respondents reported changing their behaviour (e.g. avoiding the shops) and feeling nervous about catching COVID-19 and the potential lack of social support for them and their baby (see Table 2). In terms of care, women worried about the health of their baby. They reported that they were offered virtual or telephone midwifery appointments and that support groups had moved online, however these were described as impersonal and women felt that these were less supportive than face-to-face visits. The main difficulty was in attending appointments and scans alone and the negative impact on their

**Table 1. Demographics of expectant mothers completing the survey.**

| Expectant Mothers | |
|---|---|
| **Age** | |
| Mean Age | 32 |
| **Ethnicity** | |
| White | 195 |
| Non-white | 11 |

**Table 2. Experiences of expectant mothers during the pandemic.**

| Perception of the severity of the COVID-19 pandemic |
| --- |
| *Feeling nervous about the COVID-19 pandemic* |
| "I feel nervous about being pregnant due to the pandemic (i.e. risk of catching COVID-19 but mostly lack of support from family/friends)". ***Respondent 178*** |
| "Lockdown restrictions have made me feel isolated from my family and worried about how they are able to support me when the baby arrives". ***Respondent 138*** |
| *Anxiety about contracting COVID-19* |
| "I'm also very anxious about getting COVID-19 while pregnant". ***Respondent 91*** |
| "I have stopped going to any shops for fear of COVID-19". ***Respondent 151*** |
| *Impact of COVID-19 and associated restrictions on the health and wellbeing of the baby* |
| "Being pregnant is scary but at the moment it's [a] severely stressful and emotional time which is not good for [the] baby". ***Respondent 183*** |
| "I'm nervous about being pregnant and the effects of COVID-19 on foetal development". ***Respondent 47*** |
| "I feel quite anxious, lonely and isolated with my current concerns around coronavirus and feeling me and my baby are vulnerable". ***Respondent 155*** |
| **Difference to regular appointments and delivery** |
| *Opinions on virtual appointments and services* |
| "I'm upset that I've missed out on face to face antenatal and breastfeeding classes (online classes are not the same)". *Respondent 155* |
| "Due to COVID-19 a telephone appointment was given, which feels very impersonal and not reassuring". *Respondent 1* |
| *Partner's presence at scans, appointments and delivery and impact on expectant parents' mental health* |
| "All of the restrictions have made things a lot harder and the lack of support at appointments and scans has been extremely difficult". ***Respondent 82*** |
| "My partner couldn't be with me for my scans which had an impact on both of us and our mental health". *Respondent 39* |
| "I feel extremely worried about being in labour without my partner. The worry is dominating the pregnancy". *Respondent 191* |
| "Knowing I will be admitted into hospital away from my support system is crippling me with anxiety and knowing the father cannot visit the ward after to help through the day time is worrying me for their bonding could be affected and delayed". *Respondent 4* |
| **Support from midwives** |
| *Level of contact and support received from midwives and impact on mental health and enjoyment of pregnancy* |
| "Midwife has been absolutely outstanding". ***Respondent 205*** |
| "Having good support from the midwife and mental health team has made a difference to how I feel about being pregnant". ***Respondent 125*** |
| "I don't feel I've had any support from midwives as up until I was 28 weeks pregnant I had only seen a midwife very briefly once". ***Respondent 194*** |
| "I have had no support from the midwives, I am not even sure who my midwife is". ***Respondent 72*** |
| "I haven't been able to see a midwife at all. I have had two phone calls and that's it. . .support has been non-existent". *Respondent 12* |
| "I haven't enjoyed my pregnancy as much a much as previous pregnancies. Midwife support has been fantastic but still feel very much alone". ***Respondent 2*** |
| "Midwife support has been good but I feel lonely due to not having my partner involved much". ***Respondent 11*** |
| *Communication issues* |
| "I am 25 weeks pregnant and have only met my midwife once. I have questions and concerns but no one has returned my questions or called me back. I feel very let down as I know this is not the case for expectant mothers in other trusts who have had regular contact and support from their service providers. I also work within the NHS and have adapted the way my team works and not just stopped it completely." ***Respondent 153*** |
| "When I have seen midwives their care has been great but I do feel there has been a lack of support & communication in general. Appointments have been cancelled, lacking communication about processes and updates. I feel my experience of being pregnant first time has been dampened & I haven't experienced it as I should which has led to more anxiety and less excitement". ***Respondent 90*** |

(*Continued*)

**Table 2.** (Continued)

| Perception of the severity of the COVID-19 pandemic |
| --- |
| "It feels very different to my previous pregnancy. Less contact has meant I feel less informed and less sure of my options". *Respondent 52* |
| "I was told one of my samples was going to the lab 2 weeks ago and haven't heard anything since and don't even know where I would go to receive that information". *Respondent 72* |
| "I've had to put in the work to gain context and seek guidance on the internet". *Respondent 177* |

mental health of being unable to have their partner with them, especially during labour (see Table 2). In terms of midwife support, some expectant mothers were extremely positive about the support they received from their midwife, however, many reported that the support they had received was minimal and this was the same for both mothers having their first baby and those who already had children. The level of support received was cited by participants as a key factor that influenced whether or not they had a positive experience of pregnancy. Some women reported feeling very alone during their pregnancy, including some who felt they had received good support from their midwife. Women reported that communication issues with midwives had taken some of the joy out of their pregnancy and that they felt 'left in the dark' and unsure of their options or how to find out key information.

## Total population linked data results

There were 159,620 births in Wales between 2016 and the end of 2020, with 263 removed during data cleaning due to inaccurate values for birth weight and/or gestation at birth. The remaining 159,357 babies were born to 141,679 women. Demographic details of the mother can be seen in Table 3; details of their deprivation level and living environment (rural/urban) match those reported for their babies in Table 3.

There were fewer births in 2020 than the previous average (29,031 in 2020 compared to an average of 32,582 in previous years). The population characteristics can be seen in Table 4. There was no significant difference between annual outcomes including gestation and birth weight, still birth, rates of C-section for infants born in 2020 compared to previous years. It was not possible to perform any subgroup analysis for ethnicity as ethnicity was only available for approximately 70% of births and the non-white population comprised less than 7.5% with insufficient sample size to produce any valid results.

**Table 3. Demographic details of mothers of babies in the SAIL database.**

| Characteristic | | 2016–19 | 2020 |
| --- | --- | --- | --- |
| **Age (years)** | | | |
| | **Mean** | 29.1 | 29.5 |
| | **Standard deviation** | 5.7 | 5.59 |
| | **95% CI** | 29.0–29.2 | 29.4–29.6 |
| **Ethnicity, n (%)** | | | |
| | **White** | 35,149 (91.4%) | 36,041 (91.6%) |
| | **Other** | 3,314 (8.6%) | 3,310 (8.4%) |
| **Deprivation quintile, n (%)** | **1** | 25,335 (25.7%) | 6,323 (26.3%) |
| | **2** | 22,167 (22.5%) | 5,217 (21.7%) |
| | **3–5** | 50,908 (51.7%) | 12,507 (52.0%) |
| **Home environment** | **Urban** | 98,350 (87.5%) | 21,095 (87.7%) |
| | **Rural** | 12,273 (12.5%) | 2951 (12.3%) |

**Table 4. Birth outcomes for infants born in 2020 compared to previous years.**

| Characteristic | | 2016–2019 | | 2020 | | Difference (CI) |
|---|---|---|---|---|---|---|
| | | N | (%) | N | (%) | |
| Mothers | | 113,085 | | 28,594 | | |
| Babies | | 130,326 | | 29,031 | | |
| Sex | | | | | | |
| | Female | 63,649 | (49.4%) | 14,147 | (48.7%) | - |
| | Male | 66,667 | (50.6%) | 14,879 | (51.3%) | - |
| Deprivation quintiles | | | | | | |
| | 1 | 16,125 | (24.2%) | 3,395 | (24.0%) | |
| | 2 | 13,386 | (20.1%) | 3,023 | (21.3%) | |
| | 3–5 | 37,074 | (55.7%) | 7,751 | (54.7%) | |
| Home environment | | | | | | |
| | Rural | 13,295 | (11.8%) | 3,001 | (12.3%) | |
| | Urban | 99,673 | (88.2%) | 21,425 | (88.7%) | |
| Ethnicity (baby) | | | | | | |
| | White | 83,534 | (89.5%) | 18,980 | (89.5%) | |
| | Non-white | 9,844 | (10.5%) | 2,233 | (10.5%) | |
| Still births | | 562 | (0.43%) | 104 | (0.36%) | -0.07% (-0.14% to -0.00%) |
| Gestation* | | | | | | |
| | Extremely preterm | 753 | (0.58%) | 167 | (0.58%) | - |
| | Very preterm | 1,207 | (0.93%) | 262 | (0.90%) | -0.02% (-0.05% to 0.00%) |
| | Moderate to late Preterm | 8,902 | (6.83%) | 1,907 | (6.57%) | -0.26% (-0.52% to -0.01%) |
| | Term | 114,693 | (88.00%) | 25,669 | (88.42%) | 0.41% (-0.03% to 0.86%) |
| | Late term | 4,771 | (3.66%) | 1,026 | (3.53%) | -0.13% (-0.25% to 0.00%) |
| Preterm mortality (denominator all preterm births) | | 197 | (1.89%) | 61 | (2.72%) | **0.82% (0.02% to 1.62%)** |
| Neonatal mortality (non-preterm) | | 78 | (0.07%) | 15 | (0.06%) | -0.01% (-0.02% to 0.00%) |
| Infant mortality (non-preterm) | Mortality between 29- and 90-days | 29 | (0.03%) | <5 | (0.02%) | -0.01% (-0.02% to 0.00%) |
| Birth weight** | | | | | | |
| | Extreme Low Birth Weight | 766 | (0.59%) | 166 | (0.57%) | -0.02% (-0.04% to 0.00%) |
| | Very Low Birth Weight | 988 | (0.76%) | 201 | (0.69%) | -0.07% (-0.14% to 0.00%) |
| | Low Birth Weight | 8,218 | (6.31%) | 1,875 | (6.46%) | 0.13% (0.00% to 0.26%) |
| | Normal Birth Weight | 105,612 | (81.43%) | 23,478 | (81.01%) | -0.42% (-0.76% to -0.07%) |
| | High Birth Weight | 12,140 | (9.36%) | 2,806 | (9.68%) | **0.32% (0.01% to 0.64%)** |
| | Very High Birth Weight | 1,971 | (1.51%) | 454 | (1.56%) | 0.05% (0.00% to 0.09%) |
| C-section | | | | | | |
| | Total number C-sections | 28,489 | (21.86%) | 6224 | (21.44%) | -0.42% (-1.13% to 0.29%) |
| | Elective | 1,110 | (3.90%) | 231 | (3.70%) | -0.18% (-0.37% to 0.00%) |
| | Emergency | 14,761 | (51.81%) | 3,090 | (49.49%) | **-2.17% (-4.3% to -0.04%)** |
| | Unknown | 12,618 | (44.29%) | 2,903 | (46.49%) | **2.35% (0.05% to 4.66%)** |

*Extremely preterm: <28 weeks gestation, very preterm: 28–31 weeks, preterm: 32–36 weeks, term: 37–41 weeks, late term: ≥ 42 weeks.

** Extremely low birth weight: ≤1kg, very low birth weight: 1.001–1.5kg, low birth weight: 1.501–2.5kg, normal birth weight: 2.501-4kg, high birth weight: 4.001–4.5kg, very high birth weight: >4.5kg.

**Preterm mortality.**   Although there was no evidence of an increase in preterm births, for infants that were born preterm, there appears to be a small increase in preterm mortality (see Table 4). However, this appears to reflect a temporal slow increase year on year rather than associated with an increase only in 2020 (see Fig 1).

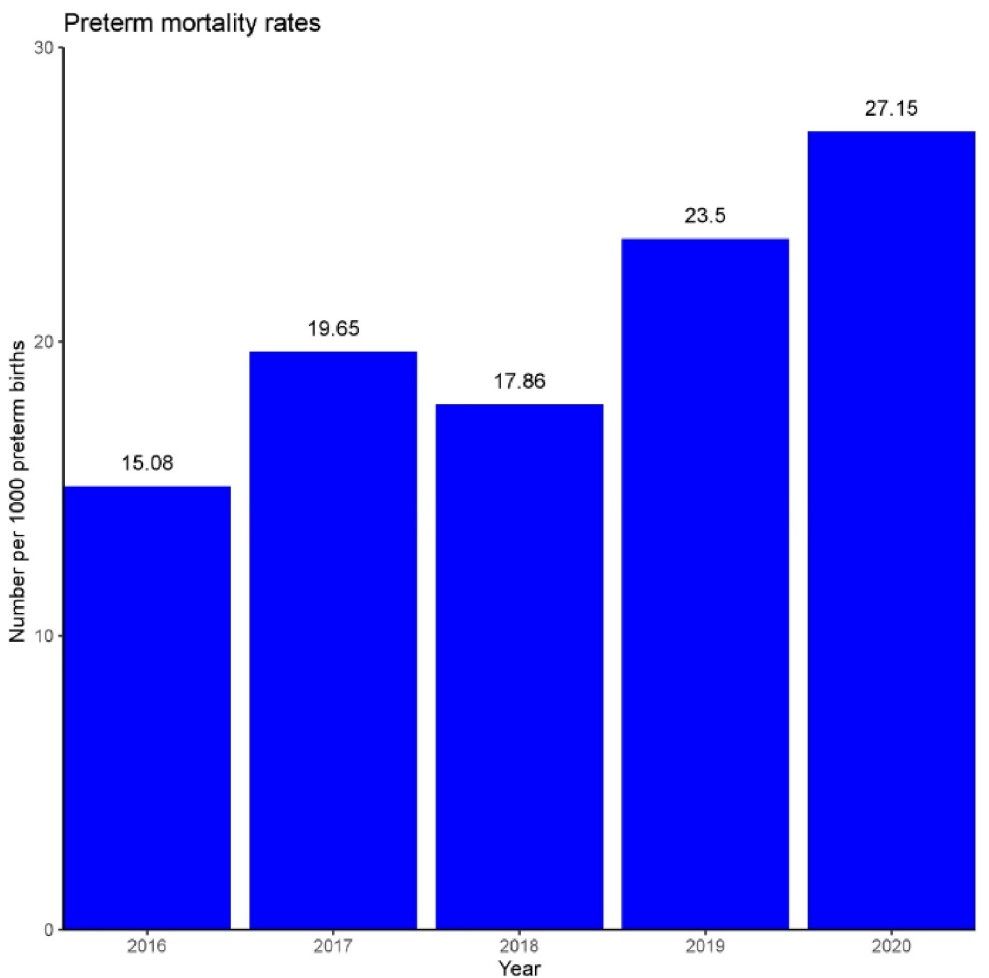

**Fig 1. Preterm mortality rates by year.** Temporal differences in mortality rates per 1000 preterm births (32–36 weeks gestation) between 2016–2020.

In consideration of this temporal trend, a post-hoc join point Poisson regression analysis was performed to test if the risk seen in 2020 was higher than predicted. There was no evidence of an increase in the risk of preterm death in 2020 (p = 0.79) when the preceding trend (2016–2019) was adjusted for.

**Caesarean sections.** The results suggest that during 2020, C-sections were more likely to be recorded without indicating if they were emergency or elective C-sections (46.5% in 2020 and previous years' mean of 44.3%). This appears to have mostly affected the emergency C-sections–the reduction in recording of emergency sections [-2.17% (-4.3% to -0.04%)] largely accounts for the increase in unknown method of C-sections [2.35% (0.05% to 4.66%)] (see Table 4).

**Birth weight.** There was a very small but not statistically significant increase in high birth weight infants born in 2020 with an odds ratio of 1.04 (95%CI: 0.99–1.08, p = 0.09).

Trends in births per month: Significant changes during the pandemic included increases in late term (≥42-weeks gestation) births in June 2020 [OR: 1.28 (95%CI: 1.04–1.58, p = 0.019)]. There was a reduction in moderate to late preterm (32-36-weeks) births in November 2020 [OR: 0.74 (95%CI: 0.61–0.89, p = 0.001)] which resulted in a reduction in prematurity overall in that month [OR: 0.79 (95%CI: 0.67–0.93, p = 0.005)].

The odds of prematurity in November were lower in 2020 compared to other years. The rate of premature birth was 6.87% in November 2020 compared to a range of 8.0% and 9.0% in previous years [OR: 0.79 (95%CI: 0.75–0.93, p = 0.005)] (see Fig 2).

Extreme preterm births in December 2020 (<28-weeks) were lower [OR: 0.27 (95%CI: 0.08–0.85, p = 0.016)] but this did not confer an overall difference in prematurity.

**Stratified by deprivation.** Late term births appear higher in the May and June less deprived group ([OR: 1.48 (CI: 1.08–2.02, p = 0.015)] and [OR: 1.72 (95%CI: 1.30–2.26, p<0.001)] respectively). In July this affected the more deprived groups only.

The reduction in November preterm births was only reflected in the most deprived quintile [OR: 0.7 (CI: 0.49–0.98, p = 0.037)] see Fig 2.

**Stratified by rural/urban.** There was no evidence of change in term births in rural areas by month (see Fig 3). In rural areas, as with the least deprived areas, there were more late term births in June during the first lockdown. In addition, preterm birth (as with trend by deprivation) were lower in November in urban areas [OR: 0.74 (95%CI: 0.61 to 0.9), p = 0.003]. Rural births showed a similar trend but were not statistically significant [OR: 0.77(95%CI: 0.44 to 1.38), p = 0.84]. Rural area preterm births increased in April during the first lockdown (11.4% in 2020 compared to 5.6% and 8.6% in previous years [OR: 1.68 (95%CI: 1.07 to 2.66), p = 0.024]) see Fig 3.

**Routine immunisations.** Ignoring the timing of the first three groups of immunisations in the childhood schedule, uptake was 100% in both 2019 and 2020. This reflects at least one of the scheduled immunisations due at 8-, 12-, and 16-weeks being given. There are some differences in the percentages that were given on-time.

Immunisations in 2020 were less likely to be given on time at 8- and 16-weeks than in 2019 (8% and 19% lower respectively). At 12-weeks the number of immunisations given on time increased to 100% in 2020 (an increase of 8%), see Table 5 and Fig 4.

## Discussion

This study found that experiencing pregnancy during the COVID-19 pandemic was stressful and difficult for the majority of the survey respondents. Expectant mothers described high levels of stress and anxiety. These results are in line with other recent studies reporting elevated stress and anxiety symptoms in pregnant women during the COVID-19 pandemic [25, 26]. From the survey responses, many expectant mothers described themselves as anxious in 2020 with the additional stress of the pandemic during their pregnancy. However, this did not translate to higher population levels of adverse events in babies. Our observational study found that although premature births were not more prevalent for the year 2020, there is evidence that they may have been higher in the first lockdown (in April) in rural areas.

In addition, the findings suggest that late term births may have been more prevalent in the first lockdown in June/July.

With the frequently changing picture of pandemic restrictions throughout 2020 and the nature of this descriptive analysis, it would be impossible to form any valid conclusions about causal relationships for any of the effects seen. One can only speculate on reasons for the outcomes observed. It is possible that disruption to healthcare services could have impacted on lengthening the gestations at delivery and urban/rural status could indicate the differences caused by proximity to such services e.g. delays to induction of labour possibly resulting in increased late term births. Other speculation could include restrictions to the lifestyles of expectant mothers–possible outcomes could be reduction of other background stressors e.g. work, appointments, day-to-day movements. Though initial anxiety at the very start of the pandemic could have caused additional anxiety prior to a period of adjustment to the new norm and an improved understanding/awareness of COVID-19.

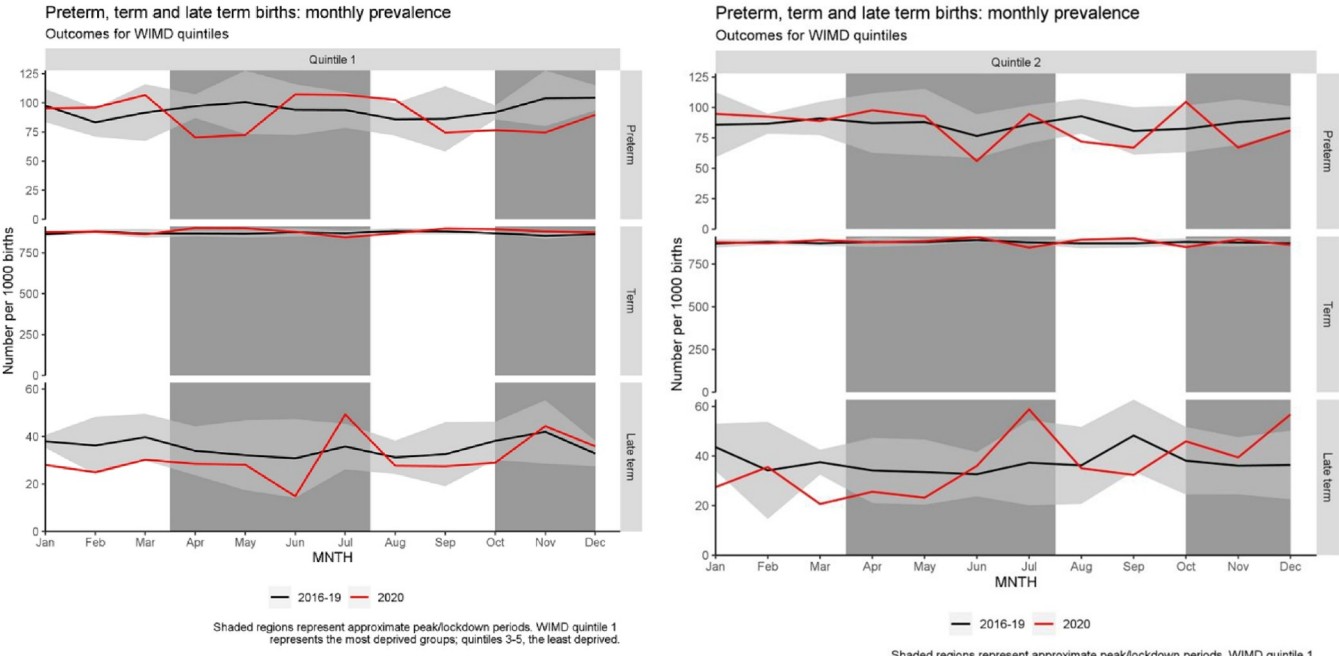

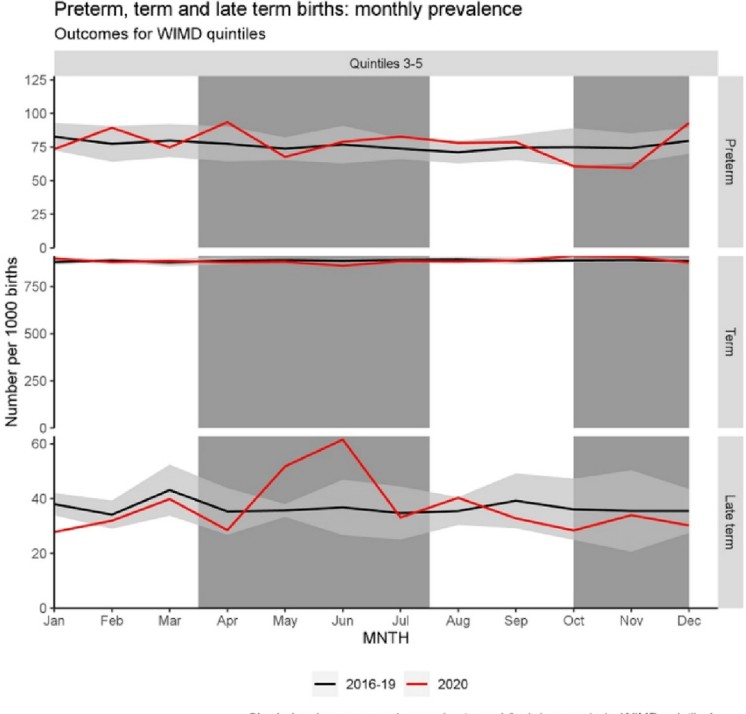

**Fig 2. Birth outcomes by month, stratified by deprivation.** Monthly prevalence of births categorised preterm (32–36 weeks gestation), term (37–41 weeks) and late-term (≥42 weeks) per 1000 births. Each represents the Welsh Index of Multiple Deprivation (WIMD) quintile to which each birth belongs. Quintile 1 represents the most deprived groups, whereas quintiles 3–5 represent the least deprived quintiles. The shaded areas in each represent approximate times during which Wales was subject to pandemic restrictions and lockdown. Black lines represent the mean number per 1000 births between 2016–2019, inclusive.

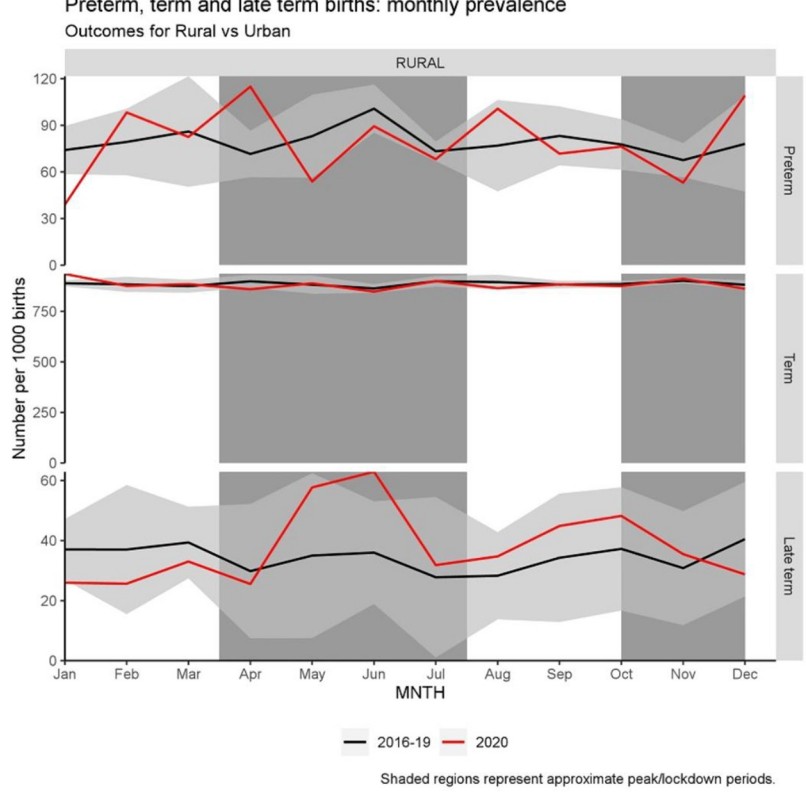

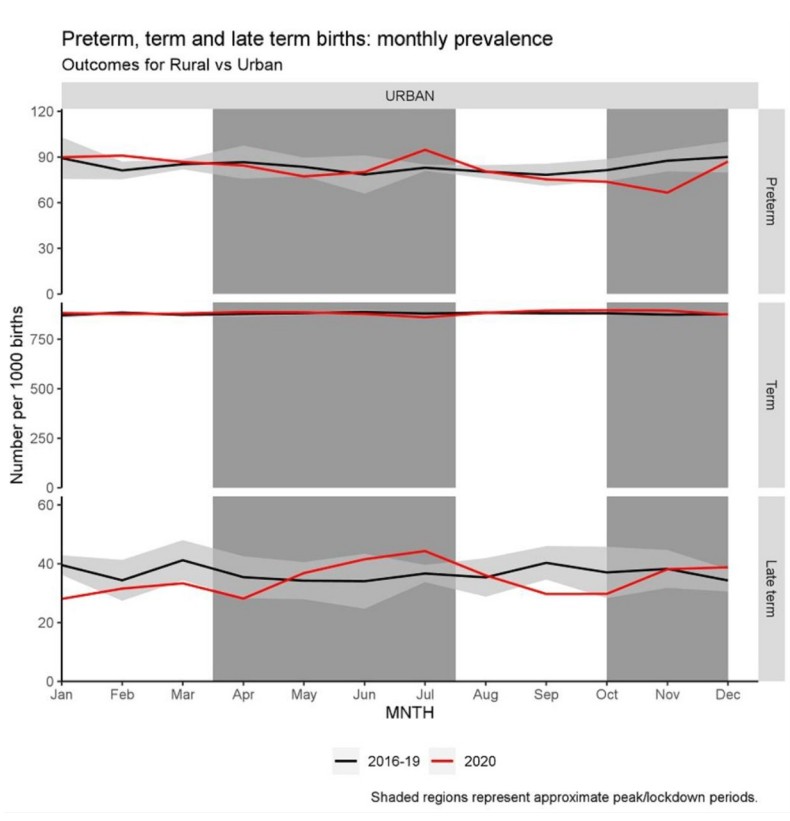

**Fig 3. Birth outcomes by month, stratified by urban/rural area.** Number of preterm (32–36 weeks gestation), term (37–41 weeks) and late-term (≥42 weeks) births per 1000 births. The first represents those born to mothers located in rural environments. The second, those in urban settings. Black lines represent the means for 2016–2019 inclusive. Shaded regions represent approximate times of peak pandemic restrictions/lockdowns in Wales.

A reduction in November 2020 urban preterm births coincided with having just completed a 'firebreak' lockdown, and a period of easing restrictions up to the Christmas period.

Mortality among preterm infants was higher in 2020 but this seems part of a temporal trend of year on year rising mortality rates. This finding is likely related to known inconsistencies across the UK as to whether extremely preterm infants are registered as a live birth [55] and increasing interventions [56] and temporal changes in the reporting of livebirths at these borderline viable gestations [57]; rather than any effect of 2020 lockdown.

At least one of the primary immunisations scheduled at each of 8-, 12-, and 16-weeks chronological age were still given to 100% of eligible babies born in 2020. (This compares to more than 95% of children under 1 year of age in Wales receiving all of their primary immunisations) [58, 59]. This suggests that the pandemic may have not made mothers more reluctant to have their infants immunised as a result of pandemic activity. Differences in the proportions of babies receiving their immunisation on time may be due to changes in the maternity and health visitor services because of the pandemic. However, Fig 4 demonstrates that, in most cases, the variation in timing of doses is largely similar between 2019 and 2020.

A study that also adopted an online survey to explore socially distanced maternity care found similar results with negative consequences of pregnancy during the pandemic including distress and emotional trauma [60]. The messaging is that pregnancy during a pandemic is a unique experience and evidence-based approaches to providing care for expectant mothers during a pandemic should be prioritised [60]. It is stressed that maternity services should establish that the provision of safe face-to-face care and access for partners or familial support are encouraged.

This study is hypothesis generating and findings will need to be confirmed by comparison with other populations and data sources such as those in the UK (England, Northern Ireland and Scotland). The authors acknowledge that the responses from participants to the online survey will be biased towards those who had access to the internet as we could not conduct face to face recruitment. The authors also recognise that there may be bias where those who were having more negative experiences were more likely to take part in the online survey, therefore overestimating the prevalence of negative experiences or equally those who were extremely distressed may not have taken part at all leading to under ascertainment. The fact that our results are consistent with other surveys does suggest that the high rates are valid although the same biases may apply to the other studies too. The survey recruited a cross-section of the population and included same sex families, ethnic minority families, younger and older mothers, those from areas of deprivation and those from non-deprived areas. There was consensus early in the study with a majority reporting a negative experience of pregnancy in lockdown.

**Table 5. Primary immunisations scheduled for 8-, 12-, and 16-weeks given on time (within 28 days of their due date).**

| Characteristic | | 2019 | 2020 | Difference |
|---|---|---|---|---|
| | | N (%) | N (%) | |
| Ave. immunisations | 8-weeks | 30,263/30,263 (100%) | 26,571/28,945 (91.8%) | -8.2% (95%CI: -7.9 to -8.5%) |
| given up to 28-days | 12-weeks | 27,837/30,259 (92.0%) | 28,942/28,942 (100%) | +8.0% (95%CI: 7.7% to 8.3%) |
| after due date | 16-weeks | 30,258/30,258 (100%) | 23,582/28,941 (81.5%) | -18.5% (95%CI: -18% to -19%) |

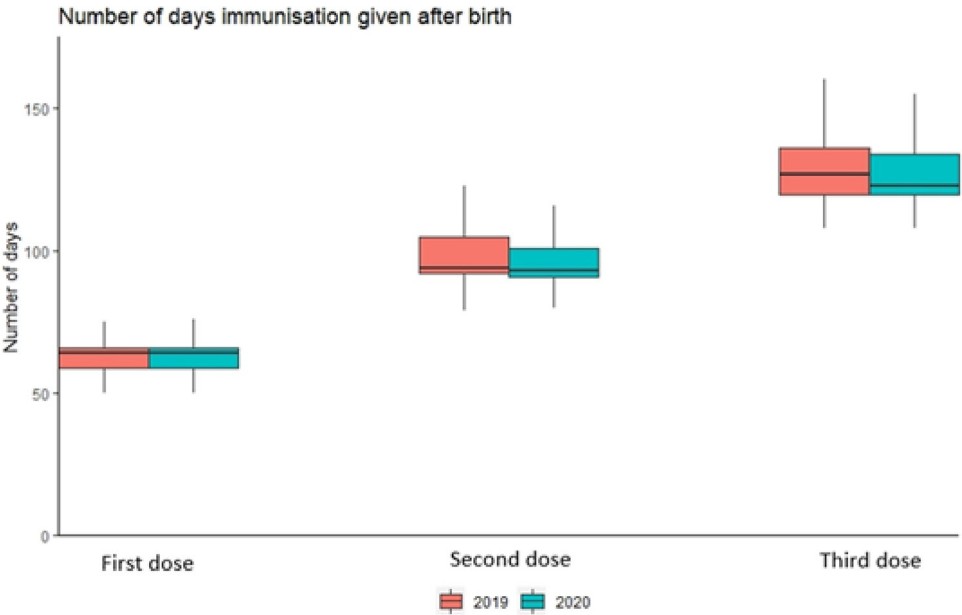

**Fig 4. Differences in timings of immunisations given in 2019 and 2020 at the recommended intervals.** The number of days after birth that infants receive their primary immunisations. The first, second and third doses relate to all immunisations that are usually due at 8, 12 and 16 weeks of chronological age.

Our findings in the population are consistent with existing similar studies, which find no clear signal that babies born during the pandemic were adversely affected. However, changes to maternity and neonatal care, as well as the direct impact of COVID-19 was different across Wales, the UK and Europe, and regional variations and impacts have been reported.

One London hospital reported a higher incidence of stillbirths during the pandemic period compared with a pre-pandemic period but no differences in births before 37-weeks gestation or caesarean delivery [30]. In contrast, a study examining national and regional data from across England found no evidence of any increase in stillbirths during the COVID-19 pandemic compared with the same period in the previous year [31]. Studies from Ireland [32], Denmark [33] and the Netherlands [34] have reported a decrease in the rate of premature births during lockdown, while a recent systematic review and meta-analysis reported: increases in stillbirth; no change in preterm births before 37-weeks gestation overall; a decrease in preterm births before 37 weeks in high-income countries; and no differences in modes of delivery, low birthweight or neonatal death [35].

The association with later term births is interesting and warrants further investigation but despite the wide-ranging changes to maternity and neonatal care that occurred during the first few months of the pandemic in Wales, the lack of measurable impact on perinatal outcomes is striking, and contrasts with broad concerns raised at the time.

In conclusion, the pandemic had a negative impact on mothers' experiences of pregnancy; however, using population-scale national data, there was little evidence that this led in general to adverse pregnancy outcomes. Lockdown periods were associated with variations in preterm (lower rates in second lockdown) and slightly higher post term births in first lockdown. There was no evidence that primary childhood immunisations uptake was lower due to lockdown measures. Infancy is usually defined as the first year of life (as relates to 'infant mortality'), and as there was insufficient data to span this period for babies born in 2020 at the time of analysis, further analysis in the first year of life and beyond will be needed to examine if stress in pregnancy has longer-term consequences for the infant and their family.

## Supporting information

**S1 Table. SAIL Databank data sources.**
(TIF)

**S2 Table. Data definitions.**
(TIF)

**S3 Table. Coding framework detailing the themes, subthemes and definitions from the qualitative analysis of women's experiences of pregnancy during the COVID-19 pandemic.**
(TIF)

**S1 Text. Software used for analysis.**
(TIF)

## Acknowledgments

This study is part the National Centre for Population Health and Wellbeing. This study makes use of anonymised data held in the Secure Anonymised Information Linkage (SAIL) Databank [43–47, 61]. We would like to acknowledge all the data providers who make anonymised data available for research.

The responsibility for the interpretation of the information supplied is the authors' alone.

## Author Contributions

**Conceptualization:** Sinead Brophy.

**Formal analysis:** Hope Jones, Mike Seaborne, Laura Cowley.

**Funding acquisition:** Sinead Brophy.

**Methodology:** Sinead Brophy.

**Project administration:** Sinead Brophy.

**Supervision:** Sinead Brophy.

**Writing – original draft:** Hope Jones, Laura Cowley.

**Writing – review & editing:** Hope Jones, Mike Seaborne, Laura Cowley, David Odd, Shantini Paranjothy, Ashley Akbari, Sinead Brophy.

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
