## [Decision Letter · Decision Letter 0]

29 Oct 2021

PONE-D-21-26751Population birth outcomes in 2020 and experiences of expectant mothers during the COVID-19 pandemic: a ‘Born in Wales’ mixed methods study using routine dataPLOS ONE

Dear Dr. Jones,

Thank you for submitting your manuscript to PLOS ONE. After careful consideration, we feel that it has merit but does not fully meet PLOS ONE’s publication criteria as it currently stands. Therefore, we invite you to submit a revised version of the manuscript that addresses the points raised during the review process.

We look forward to receiving your revised manuscript.

Kind regards,

Maria Christine Magnus, MPH

Academic Editor

PLOS ONE

Journal Requirements:

2. When reporting the results of qualitative research, we suggest consulting the COREQ guidelines  or other relevant checklists listed by the Equator Network, such as the SRQR, to ensure complete reporting (http://journals.plos.org/plosone/s/submission-guidelines#loc-qualitative-research). Moreover, please provide the interview guide used as a Supplementary File.

3. You indicated that you had ethical approval for your study. In your Methods section, please ensure you have also stated whether you obtained consent from parents or guardians of the minors included in the study or whether the research ethics committee or IRB specifically waived the need for their consent.

4. Please amend your current ethics statement to address the following concerns:  

a) Did participants provide their written or verbal informed consent to participate in this study? 

This study is part the National Centre for Population Health and Wellbeing, which is funded by Health Care Research Wales. 

7. We note that you have indicated that data from this study are available upon request. PLOS only allows data to be available upon request if there are legal or ethical restrictions on sharing data publicly. For more information on unacceptable data access restrictions, please see http://journals.plos.org/plosone/s/data-availability#loc-unacceptable-data-access-restrictions. 

8. Please note that in order to use the direct billing option the corresponding author must be affiliated with the chosen institute. Please either amend your manuscript to change the affiliation or corresponding author, or email us at plosone@plos.org with a request to remove this option.

Additional Editor Comments:

We have received comments from two reviewers on your paper who have expressed some concerns about the methods used. I ask you to address their comments carefully before I can make a final recommendation.

Reviewers' comments:

Reviewer's Responses to Questions

**Comments to the Author**

1. Is the manuscript technically sound, and do the data support the conclusions?

Reviewer #1: Yes

Reviewer #2: Partly

2. Has the statistical analysis been performed appropriately and rigorously? 

Reviewer #1: Yes

Reviewer #2: No

3. Have the authors made all data underlying the findings in their manuscript fully available?

Reviewer #1: No

Reviewer #2: No

4. Is the manuscript presented in an intelligible fashion and written in standard English?

Reviewer #1: Yes

Reviewer #2: No

5. Review Comments to the Author

Reviewer #1: This is a mixed-methods study with quantitative evaluation of perinatal outcomes (including analysis stratified by rural/urban geography, deprivation indices, gestational age and birthweight) and childhood immunization delays during the pandemic, as well as qualitative evaluation of maternal stress and anxiety. Inclusion of evaluation of childhood immunizations and maternal stress/anxiety is novel and of great interest compared to prior studies published mostly on perinatal outcomes during the pandemic vs. pre-pandemic epochs. I was hoping there would be additional analysis performed connecting maternal/stress anxiety (could code as categorical variables) & association with maternal outcomes among the 215 respondents surveyed, but recognize the small “n” would limit statistical significance and power of this analysis compared the large sample size including in the initial quantitative analysis of nearly 30,00 women.

Abstract/Introduction

•In background (both in abstract and page 5 of introduction), suggest specifying primary childhood immunisations.

Methods

•State 95% confidence intervals used with p-value used to determine significance (presumable 0.05).

•Include description of post-hoc Poission analysis in methods

•Page 8, Deprivation Quintiles: What socioeconomic factors are included with this deprivation index?

•Consider analysis of maternal/stress anxiety & association with adverse perinatal outcomes

Results

•Page 12: was December 2020 still a lockdown period in Wales?

•Do the authors have information on sociodemographic factors by which subgroup analyses could be performed? Recommend providing a table of baseline study characteristic including patient demographics for both the larger ~29,000 patients and 215 patients with qualitative analysis

Discussion

•Consider additional discussion of results related to rural/urban analysis and deprivation indices. Do the authors propose potential hypotheses for their findings of reduced preterm birth in the higher deprivation group for November 2020?

Table

•Table 2: specify definitions for extreme preterm, very preterm, moderate-late preterm etc. either in the methods or as a footnote; specify definitions for extreme low birth weight, low birth weight, high birth weight, etc.

•Re-label Tables 1, 2, 3 in the order in which they appear. I believe Table 1 with qualitative data should be Table 3 as it appears last

•Figure 2 showing longitudinal time course of preterm births my months in line graph form is new compared to other studies and helpful presentation of temporal trend.

Reviewer #2: This paper covers several important topics; to investigate birth outcomes in 2020 compared to 2016-2019, experiences of expectant mothers during the COVID-19 pandemic as well as how immunisations have been provided during the pandemic compared to the previous year.

However, my major concern regards the broad span of research questions that are addressed. I would suggest the authors to either focus on the birth outcomes, the qualitative part of the paper with expectant mothers’ experiences or the immunisations.

I believe that this manuscript may be of interest for the readers of PONE, but I have some suggestions necessary to improve the manuscript:

Introduction

Due to the fact that the paper is aiming to cover a number of research questions, I found that the introduction is somewhat straggling and incoherent. At the same time there is no mention of immunisations in the introduction.

In the aim the authors mentions “vaccination rates” which the readers may interpret as COVID-19 vaccination rates. I assume that the authors here imply routine immunsations in childhood? Please clarify.

Minor:

•Page 4, para 1, row 3: How does “Maternal stress and anxiety during

pregnancy is also known to be associated with adverse neonatal and obstetric outcomes [7,8,9]” differ from “The relationship between prenatal anxiety and obstetric outcomes is less clear [19,20]”. This is a bit unclear to me, and could either be shortened or explained better.

•Page 4, para 2: missing a dot after reference 22.

•Page 5, para 1: add “2020” after March for clarification.

•Page 5, para 1: Please rephrase the sentence starting with “September to December increased…” as I don’t think the months actually increased but potentially the cases.

•Page 6, para 1: missing a dot after reference 38.

Methods

Study design part 1): The authors state that the data is “routinely available”. What does this mean? Can anyone use the data? Please move the section on “total population linked data” to the beginning of the methods for coherency as these results are presented before the survey online.

The final sentences on page 7, last para, are unclear and need to be clarified and revised: “Missing data was treated as missing except in the case of missing gestational age and, therefore, gestational age category (missing in 1327 [0.8%] cases). As the majority of the birth weights in these cases were consistent with term weights, these were assigned as 40-weeks gestation and ‘term’ category respectively.”

Using this type of design, comparing the years 2016-2019 with 2020, how can you know that the potential difference in outcomes is related to the pandemic? What about temporal or seasonal trends in general? Would it be possible to use another approach such as difference-in-difference model as suggested by Been JV, et al. Impact of COVID-19 mitigation measures on the incidence of preterm birth: a national quasi-experimental study. The Lancet Public Health. 2020;5(11):e604-e11.

The same thing goes for childhood immunizations. How can we know that potential differences is due to COVID-19 measures and not just a coincidence or trend that was initiated before the pandemic?

In addition, on page 8, para 5, it states that “all babies have received recommended immunsations”, does this mean that the coverage of routine immunsations in childhood in Wales is 100%? This would of course be amazing, but is it really correct? Is this the same with the following doses?

How did you decide on 28-days for to be “on time”?

Study design part 2: Do you have any knowledge about how those responding to the online survey differed from those not participating in the online survey? For instance, which demographic characteristics were collected and could these be compared with other pregnant women? This should be deliberated in more depth in the discussion.

Results

As the manuscript covers several outcomes, a lot of results are being presented and there is a need to structure the text somewhat for the reader to follow. I also suggest to reduce the number of outcomes and restrict the paper to one method. In addition, it seems like some of the statistical analyses that are mentioned in the results section is not part of the method section? Eg. the post hoc join point Poisson regression.

Minor:

•Make sure that the tables are described in the right order. Now table 2 and 3 comes before table 1. Either revise the order of the tables or change the numbering of the tables so that the birth outcomes are table 1 and so forth.

•Only present on decimal for percentages in the tables.

•There is no need to present 3 decimals for non-significant p-values (eg page 7; p=0.785)

•Page 11, para 2: Are these results possible to see somewhere? Which table is the text referring to and how does the results imply this?

•Page 11, para 3: a dot is missing in the p-value, and a “s” after odd(s) ratio.

•Page 12: Should “stratified by rural/urban” be a subheading?

•There is an inconsistency in the use of small and capital letters in “Figure/figure/Table/table”.

•Table 3: there are some missing commas (eg. 30263). Furthermore, in the text to the table on page 13 it states that uptake of immunisations was 100% in both 2019 and 2020. In the table it looks like 91.8% were vaccinated. It should be clarified which the source population is and if it is correct that all babies born in 2019 were vaccinated.

•Some of the sentences are started with a percentage, eg “25%” on page 14. I would suggest to revise these sentences or write out the number in letters.

Discussion

What do the authors believe is the reason for the results of increased numbers of late term births mean? Chance finding? Reduced number of inductions due to lack of staff? Please elaborate.

On page 17 it is stated that “This finding is likely related to increased interventions in preterm births and thus recording them as live births rather than stillbirths [52];

rather than any effect of 2020 lockdown.”. What does this mean? Are there live births that are mistakenly recorded as stillbirths?

On page 19, first row it is stated “There was consensus early in the study with a majority reporting a negative experience of pregnancy in lockdown.” What does this imply? Please clarify.

The major limitation I see is the possibility of temporal and even seasonal trends in preterm birth for instance and I suggest that authors to further investigate the possibility of seasonal trends being the reason for their findings.

Minor:

•Page 18, first para, last row: “is” should be replaced with “in”?

Conclusion

The conclusion states that further analysis in the first year of life will be needed to examine if stress in pregnancy has longer-term consequences for the infant and their family. I would think that analyses of first year of life won’t be enough for longer-term consequences.

6. PLOS authors have the option to publish the peer review history of their article (what does this mean?). If published, this will include your full peer review and any attached files.

Reviewer #1: No

Reviewer #2: No

---

## [Author Response · Author response to Decision Letter 0]

21 Jan 2022

Thank you for the comments and all requested changes have been made and are noted in the Response to Reviewers document. 

1. File names have been amended to meet the requirements

2. The survey questions are attached in a file called New Expectant Mum Questionnaire

3. Consent information added to methods section

4. The ethics statement has been amended with information about consent

5. Funding information has been updated

6. Funding statement has been updated

7. Data Accessibility statement has been amended

8. The corresponding author is affiliated with the chosen institute. 

9. We have updated the captions for the supporting information

---

## [Decision Letter · Decision Letter 1]

14 Mar 2022

PONE-D-21-26751R1Population birth outcomes in 2020 and experiences of expectant mothers during the COVID-19 pandemic: a ‘Born in Wales’ mixed methods study using routine dataPLOS ONE

Dear Dr. Jones,

Thank you for submitting your manuscript to PLOS ONE. After careful consideration, we feel that it has merit but does not fully meet PLOS ONE’s publication criteria as it currently stands. Therefore, we invite you to submit a revised version of the manuscript that addresses the points raised during the review process.

We look forward to receiving your revised manuscript.

Kind regards,

Maria Christine Magnus, PhD

Academic Editor

PLOS ONE

Journal Requirements:

Additional Editor Comments:

Thank you for submitting your revised manuscript. The reviewers have some additional comments that I would like you to address. I look forward to receiving your revised manuscript.

Reviewers' comments:

Reviewer's Responses to Questions

**Comments to the Author**

1. If the authors have adequately addressed your comments raised in a previous round of review and you feel that this manuscript is now acceptable for publication, you may indicate that here to bypass the “Comments to the Author” section, enter your conflict of interest statement in the “Confidential to Editor” section, and submit your "Accept" recommendation.

Reviewer #1: (No Response)

Reviewer #2: All comments have been addressed

2. Is the manuscript technically sound, and do the data support the conclusions?

Reviewer #1: Yes

Reviewer #2: Yes

3. Has the statistical analysis been performed appropriately and rigorously? 

Reviewer #1: Yes

Reviewer #2: Yes

4. Have the authors made all data underlying the findings in their manuscript fully available?

Reviewer #1: No

Reviewer #2: Yes

5. Is the manuscript presented in an intelligible fashion and written in standard English?

Reviewer #1: No

Reviewer #2: Yes

6. Review Comments to the Author

Reviewer #1: • Suggest reference to “pregnant persons” instead of pregnant “women” throughout the manuscript to increase inclusivity.

• The introduction includes hardly any reference to childhood immunizations, nor relevant tie in to the novelty proposed by the authors in the introduction (and discussion) that we need to better understand how stress among pregnant persons is associated with this change in birth outcomes during the pandemic. I would recommend removing any research/data related to immunizations from this manuscript as this topic seems it could and should be its own separate manuscript. It is too much to have three separate research questions (birth outcomes, immunizations, stress/experiences) in one manuscript, and therefore leads to a lack of focus. This recommended change by a reviewer was not adequately addressed in this revision.

• Also, I recommend presenting the survey data regarding birthing people’s experiences (this is the exposure of interest assessed which should precede the outcomes of interest i.e. the birth outcomes), then outcome data reported second.

• Can some of the introduction be moved to the conclusion? Typically introduction should be kept to 1-1.5 pages and present a succinct, clear rationale for the manuscript with relevant background. The introduction here can be tightened up to include only the most pertinent details.

• Line 135—remove additional period after “missing”

• Line 142: It is inappropriate phrasing to compare outcomes to a year (2020). Specify that you are comparing pre-pandemic outcomes to outcomes in the 2020 pandemic epoch.

• Line 267: Add bracket in front of OR.

Reviewer #2: (No Response)

7. PLOS authors have the option to publish the peer review history of their article (what does this mean?). If published, this will include your full peer review and any attached files.

Reviewer #1: No

Reviewer #2: No

---

## [Author Response · Author response to Decision Letter 1]

1 Apr 2022

Reviewer #1 Comments: 

• Suggest reference to “pregnant persons” instead of pregnant “women” throughout the manuscript to increase inclusivity.

Thank you for this comment and your feedback. However, we feel that changing the terminology to “pregnant persons” may be unclear or confusing for readers who may not know if this relates to the one who is pregnant or the partner as they are also expecting. After discussions with midwives, they agree that pregnant women or expectant mum is the appropriate terminology. 

• The introduction includes hardly any reference to childhood immunizations, nor relevant tie in to the novelty proposed by the authors in the introduction (and discussion) that we need to better understand how stress among pregnant persons is associated with this change in birth outcomes during the pandemic. I would recommend removing any research/data related to immunizations from this manuscript as this topic seems it could and should be its own separate manuscript. It is too much to have three separate research questions (birth outcomes, immunizations, stress/experiences) in one manuscript, and therefore leads to a lack of focus. This recommended change by a reviewer was not adequately addressed in this revision.

Thank you for this comment. The introduction states “there are a lack of studies examining experiences of pregnancy during the pandemic in combination with national data on objective measures of obstetric, neonatal and infant outcomes including immunisation uptake which may potentially indicate disruption to usual access to healthcare” and we feel this provides a justification for investigating immunisations in this piece of work. We believe this research would lose something by removing all information about immunisations during the pandemic. We believe the research on immunisations is important and should be in the public domain and is fundamentally linked to the experiences and outcomes for pregnant women through COVID.

• Also, I recommend presenting the survey data regarding birthing people’s experiences (this is the exposure of interest assessed which should precede the outcomes of interest i.e. the birth outcomes), then outcome data reported second.

This has been amended so the survey data is presented first in the methods and results

• Can some of the introduction be moved to the conclusion? Typically introduction should be kept to 1-1.5 pages and present a succinct, clear rationale for the manuscript with relevant background. The introduction here can be tightened up to include only the most pertinent details.

Some of the introduction has been removed to make it more succinct and clearer

• Line 135—remove additional period after “missing”

This has been amended

• Line 142: It is inappropriate phrasing to compare outcomes to a year (2020). Specify that you are comparing pre-pandemic outcomes to outcomes in the 2020 pandemic epoch.

The wording has been changed here

• Line 267: Add bracket in front of OR. 

This has been amended

---

## [Editor Report · Decision Letter 2]

5 Apr 2022

Population birth outcomes in 2020 and experiences of expectant mothers during the COVID-19 pandemic: a ‘Born in Wales’ mixed methods study using routine data

PONE-D-21-26751R2

Dear Dr. Jones,

We’re pleased to inform you that your manuscript has been judged scientifically suitable for publication and will be formally accepted for publication once it meets all outstanding technical requirements.

Kind regards,

Maria Christine Magnus, PhD

Academic Editor

PLOS ONE

---

## [Editor Report · Acceptance letter]

9 May 2022

PONE-D-21-26751R2 

Population birth outcomes in 2020 and experiences of expectant mothers during the COVID-19 pandemic: a ‘Born in Wales’ mixed methods study using routine data 

Dear Dr. Jones:

I'm pleased to inform you that your manuscript has been deemed suitable for publication in PLOS ONE. Congratulations! Your manuscript is now with our production department. 

Kind regards, 

on behalf of

Dr. Maria Christine Magnus 

Academic Editor

PLOS ONE